# Abundance of the vector *Aedes aegypti* in urban and rural areas in Managua, Nicaragua

Harold Suazo-Laguna[1], Jacqueline Mojica-Díaz[1], María M. Lopez[1], Angel Balmaseda[1], Eva Harris[2], Josefina Coloma[2], Jose G. Juarez [1,3]*

1 Sustainable Sciences Institute, Managua, Nicaragua, 2 Division of Infectious Diseases and Vaccinology, School of Public Health, University of California, Berkeley, California, United States of America, 3 College of Veterinary Medicine and Biomedical Sciences, Texas A&M University, College Station, Texas, United States of America

* jjuarez@icsnicaragua.org, jua05396@tamu.edu

## Abstract

### Background

*Ae. aegypti* is the primary vector of dengue, chikungunya, Zika, and yellow fever viruses, traditionally associated with urban environments. However, its presence and abundance in rural settings remain understudied. This study compares *Ae. aegypti* populations between rural and urban communities in Managua, Nicaragua, across different seasons over multiple years.

### Methods

Entomological surveys were conducted in 500 randomly selected houses (250 rural, 250 urban) during the rainy and dry seasons of 2022 and 2023. Immature mosquitoes were collected from water-holding containers, and adult mosquitoes were sampled using aspirators. *Stegomyia*, pupal, and adult indices were calculated. Generalized linear mixed models (GLMM) and generalized additive mixed models (GAMM) were used to evaluate factors associated with pupal productivity and adult female abundance.

### Results

Rural communities consistently exhibited higher immature and adult *Ae. aegypti* abundance than urban communities across most entomological indices. Rural households accounted for 74% of collected pupae and had 3.34 times higher pupal counts compared with urban households (95% CI: 2.17–5.15, p < 0.001). Barrels were the most productive container type in both settings, with uncovered barrels increasing pupal counts by 73% (IRR = 1.73, 95% CI: 1.63–1.82, p < 0.001). Total adult female abundance was significantly higher in rural communities compared to urban ones (IRR = 1.39, 95% CI = 1.13 − 1.72, *p* < 0.001). Rural households also exhibited 67%

**Data availability statement:** Data set is available in ZENODO: https://doi.org/10.5281/zenodo.18882446 and R code can be found at: https://github.com/jgjuarez/UrbanRuralManagua.

**Funding:** This study was funded by the National Institute of Allergy and Infectious Diseases of the National Institutes of Health, grant U01 AI151788 (EH, JC), as part of the Centers for Research on Emerging Infectious Diseases (CREID) network. The funders had no role in study design, data collection and analysis, decision to publish, or preparation of the manuscript.

**Competing interests:** The authors have declared that no competing interests exist.

higher female rates per person than urban households (IRR = 1.67, 95% CI: 1.33–2.09). Nonlinear modeling revealed threshold-like dynamics between pupal counts and adult female abundance, with sharp increases in adult density once pupal abundance exceeded 50 pupae.

## Conclusion

We consistently observed that rural communities in Managua sustain significantly higher *Ae. aegypti* populations than urban ones. The identification of pupal thresholds linked to adult abundance highlights the importance of preventing containers from reaching high productivity levels. Overall, these findings challenge the perception of *Ae. aegypti* as predominantly urban and underscore the need for integrated rural–urban surveillance and context-specific vector control strategies.

## Author summary

Mosquitoes continue to spread to new areas around the world, carrying viruses that cause diseases such as dengue, chikungunya, Zika, and yellow fever, which cause a wide range of symptoms from mild fevers to fatal outcomes. The main mosquito responsible for transmitting these diseases, *Aedes aegypti*, has traditionally been considered an urban species, thriving in cities where water storage and human activity create ideal larval development sites. However, little is known about its abundance in rural communities. In this study, we examined mosquito populations in urban and rural areas of Managua, Nicaragua, during both rainy and dry seasons of 2022 and 2023. We found that *Ae. aegypti* was more abundant in rural communities across multiple entomological indices. We also found that uncovered water storage barrels were strongly associated with increased pupae production. Our findings challenge the long-standing perception of *Ae. aegypti* as an urban mosquito and emphasize the need to include rural areas in vector surveillance and control programs. Understanding these patterns is critical for designing more effective strategies to prevent mosquito-borne diseases in diverse environments.

## Background

Arthropod-borne viruses (arboviruses) such as dengue, chikungunya, Zika, and yellow fever viruses are the most important human viral diseases transmitted by mosquitoes [1]. The public health impact of these diseases has increased over the past decades, especially due to dengue virus (DENV), which has spread to new geographic locations and places more than 4 billion people at risk of infection globally [2]. These arboviruses are transmitted by mosquitoes of the genus *Aedes*, with *Ae. aegypti* (L.) being the main vector for disease transmission [3]. *Ae. aegypti* is a highly anthropophilic mosquito, with its global distribution expanding due to the effects of

climate change [4]. As such, it is critical to fully understand how local populations of *Ae. aegypti* are modulated by spatio-temporal factors such as location and seasonality over time [5–7]. However, several regions of Central America remain with scarce publications regarding location and seasonality data for *Ae. aegypti*.

In Nicaragua, dengue has been the most significant mosquito-borne disease since 1985, when the first epidemic was recorded [8]. Over the past decades, DENV has caused epidemics every 2–3 years [9–11]. Previous work in Managua has shown the potential of improving ongoing vector control of *Ae. aegypti* by means of community engagement and source reduction [12–14]. These studies have also provided information regarding productive containers for immature stages, seasonal trends, and risk factors that modulate local urban populations of *Ae. aegypti*. In these urban communities, we have previously observed the importance of water storage practices, waste management, and larvicides modulation of entomological indices [12,15]. Additionally, urban mosquito productivity has been associated with factors such as container type, environmental conditions, climate, and seasonality [5,16,17]. Most studies in Nicaragua and elsewhere have focused on *Ae. aegypti* populations found in urban environments, largely because this mosquito is commonly associated with urban settings. It is thought that urbanicity issues such as dilapidated infrastructure, erratic supply of water or water intermittency, poor sanitation services, and increasing human population density can create ideal conditions for high *Aedes* mosquito densities. However, the importance of *Ae. aegypti* and the diseases it transmits has also been demonstrated in rural areas [18–21]. Nonetheless, limited information is currently available for seasonal mosquito abundance patterns in rural communities in Central America. Other rural areas have observed that density of children, lot size, water containers, and water storage practices impact mosquito abundance in rural areas [18–21]. Notwithstanding, the definition of what is "urban" or "rural", and where urbanicity starts and finishes can vary in both time and place, making comparison difficult. Further, it is critical to compare areas where entomological measurements are performed concurrently within a well-defined spatiotemporal framework.

The objective of this study was to determine how *Ae. aegypti* entomological indices differ in urban and rural areas and identify which larval development/oviposition containers were associated with higher productivity by area. We found that all entomological indices of *Ae. aegypti* were higher in "rural" as compared to "urban" areas in both dry and rainy seasons across multiple years, challenging the concept of *Ae. aegypti* as an urban mosquito.

## Materials and methods

### Ethics statement

Protocols for the entomological surveillance and socio-demographic questionnaires were reviewed and approved by the Institutional Review Boards (IRB) of the University of California, Berkeley (A2CARES: 2021-03-14191) and the Nicaraguan Ministry of Health (A2CARES: CIRE 02/08/21–114 Ver.4). Residents provided signed consent during the initial visit and verbal consent for each subsequent visit.

### Study area

This study was conducted in District III of Managua, Nicaragua (Fig 1), during the dry (February-March) and rainy (October-November) seasons of 2022 and 2023. The capital city of Managua comprises seven districts, of which District III is the largest, with an area of 73.2 km$^2$ and a population of ~187,000 inhabitants. It is located 100–400 meters above sea level, with a predominantly tropical climate characterized by two seasons (dry: December-April; rainy: May-November) [22]. In 2022, the mean surface air temperature was 25.8°C with 2,438 mm of precipitation, while in 2023, it was 26.2°C with 2,234 mm of precipitation [23]. We worked in 21 neighborhoods of District III, within the catchment area of two health posts in Camilo Ortega (12°05'47.4"N, 86°18'22.7"W) and Nejapa (12° 6'45.56"N, 86° 19'27.78"W). Camilo Ortega is designated as an urban area with ~26,000 inhabitants, and Nejapa is referred to as a rural area, with ~11,000 inhabitants. We used the official classification system of the Nicaraguan Ministry of Health (MOH) to classify urbanicity within our communities [24].

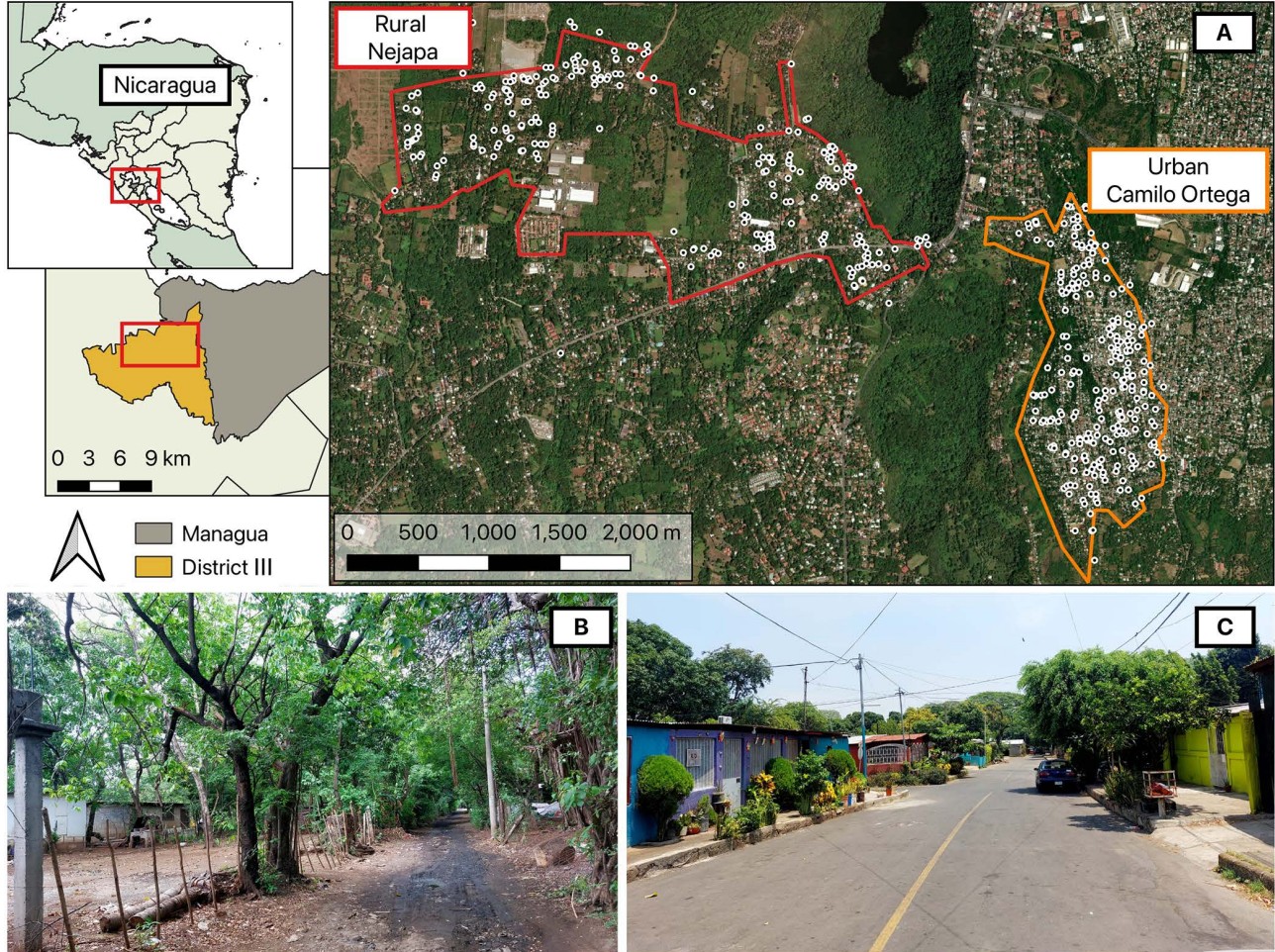

**Fig 1. Entomological collections conducted in dry and rainy seasons of 2022-2023 in communities of District III of Managua, Nicaragua. A)** Rural (Nejapa) and Urban (Camilo Ortega) communities of District III. Black dots indicate the approximate distribution of study households. Geographic coordinates were randomly offset to obscure exact household locations. **B)** Street view of a typical household structure and adjacent road in Nejapa. **C)** Street view of typical household structures and adjacent road in Camilo Ortega. Map was generated using Quantum GIS (QGIS 3.30) using freely available administrative boundaries and map data from OpenStreetMap [65] for satellite imagery.

In Nicaragua, the definition of "urban" is any locality that is a departmental, regional, or municipal capital, or any population center with ≥1,000 inhabitants that also has basic urban infrastructure (i.e., laid-out street grid, electricity supply, and commercial or industrial establishments). Rural areas are defined by the exclusion of the above and dispersed populations living outside defined settlements [25].

### Study design and sample size

The entomological surveillance program established was part of a community-based cohort study evaluating arboviruses. The parent cohort had > 2,000 participants of ages 2–80 years, living in ~1,000 houses (~500 each in urban and rural communities) who underwent a yearly evaluation of arboviral infections and diseases. Participants with febrile episodes also visited the study health post within their community for arbovirus disease testing. For the study presented here, a sample size of 468 houses (234 each in Camilo Ortega and Nejapa) was estimated based on a 20% house infestation rate during

the dry season, at 95% confidence with a margin of error of 5%. We assumed a 5% non-response rate due to community members not being available, thus targeting a total sample size of 500 houses (250 per community). These 500 houses, with a minimum 50m-radius between houses, were randomly selected from the parent cohort of ~1,000 houses for entomological surveillance. The parent cohort house selection was performed using a framework for patches of risk based on landscape features (*i.e.*, vegetation, house rooftops, and open areas). To create a spatially representative sampling design, we utilized satellite imagery of the rural and urban study areas in District III, using impervious surfaces (man-made structures) as a proxy for inhabited sampling locations, which were then selected randomly, taking into account close-pair points and allowances for misclassification and inaccessibility. Houses from the previous visits that were found closed or un-inhabited were replaced with the closest house from the parent cohort to maintain 500 houses per survey. A total of 384 (77%) houses had the complete set of 4 measurements. Of the 116 (23%) houses that were replaced, the most common reason for drop-out was participant migration (>60%).

**Entomological surveillance**

Field visits were conducted from 8 am to 5 pm Monday to Friday, with occasional Saturday visits for houses that were not available during the weekdays. As part of standard procedures, the MOH also conducted routine *Ae. aegypti* control and surveillance activities across all districts of Managua, which could have affected some of the study households. Every two months, MOH vector control field technicians visited houses to eliminate immature vectors by placing larvicide in clean water containers in the house. Complementary activities carried out included pesticide fumigation and community clean-up campaigns. To minimize interference, we scheduled our activities to avoid overlapping with MOH vector control efforts.

Immature mosquito collections. Larvae and pupae were collected during house visits by study personnel, accompanied by a resident. Inspections began outdoors and concluded indoors, with field technicians moving systematically from right to left to cover all areas. Each water-holding container was examined, and the following characteristics were recorded: type, location (inside or outside), use of water (*i.e.*, consumption, human use, cleaning, irrigation, nothing), frequency of scrubbing, condition of covering (full, half, or none), material, presence of moss on the walls, and visible presence of larvicide. Containers were classified into five categories: 1) barrels (volume of ≥200 liters of water), 2) cement laundry sinks, 3) tires, 4) buckets (volume of ≤20 liters), and 5) other containers (flower pots, jugs, animal water dishes, pieces of plastic, among others). Larvae and pupae found were collected using a net, a Pasteur pipette, and a small plastic bowel. The procedure lasted 15–20 minutes, depending on the property size and number of containers. Immature stages were stored in jars with 70% alcohol, labeled with the house unique identifier, and transported to our laboratory facilities. Taxonomic identification of 4th instar larvae and pupae was conducted for *Ae. aegypti* and *Ae. albopictus* by a trained technician, with all immature stages counted and included in the analysis. Identification followed dichotomous keys [26–29], with quality control verified by our lead medical entomologist. All specimens were identified and counted using a QZE Zoom Stereo binocular microscope.

Adult collection. Adult mosquitoes were collected using Prokopack aspirators, with sampling also starting outdoors and moving indoors. The outdoor aspiration procedure was carried out on the exterior walls of the house focusing on plants, water barrels, tires, and animal water dishes. Indoor aspirations were thorough, with all living quarters inspected (bedrooms, living room, kitchen and bathrooms), sampling under, behind and over furniture, beds and closets. Different collection cups were used for indoor and outdoor collections, labeled with the house unique identifier, and stored in a thermal bag carried by the field worker, maintaining a temperature of 4–8°C to avoid damage to the mosquitoes.

All mosquitoes were transported to the laboratory field station and stored at -20°C until further processing. Mosquitoes were separated by species (*Ae. aegypti, Ae. albopictus, Culex quinquefasciatus, Culex coronator, Anopheles albimanus*, others) [29,30], sex (male and female), and engorged state (bloodfed and unfed). This adult aspiration procedure lasted between 10–15 minutes per collection site.

## Entomological indices

For each entomological visit, we calculated the *Stegomyia* indices (House index, Container index, and Breteau index), pupal indices (pupae per person, pupae per container, and pupae per house), and adult indices (adult index, adults per person, adults per house, and females per person) (Table 1) [31]. Container productivity was also estimated as the percentage of total *Ae. aegypti* pupae collected by type of container [32–34].

## Data analysis

Initial descriptive analyses were performed for data exploration. Container productivity analysis was evaluated for *Ae. aegypti* pupae, as it has been shown that this entomological indicator is the most accurate to translate to adult abundance [33,35]. Entomological indices were descriptively summarized by season (dry-rainy), year (2022–2023) and community type (urban-rural). During the initial exploratory analysis, we identified a single observation in the pupal count data with a standardized z-score greater than 20, indicating an extreme value relative to the overall distribution. In general, we avoid removing outliers to preserve biologically plausible variation in mosquito productivity. However, diagnostic evaluation indicated that this observation exerted disproportionate leverage on the mixed-effects model estimates, substantially influencing model convergence and parameter interpretation. Therefore, this observation was removed from the modeling dataset to prevent undue influence on the results and to improve the stability and interpretability of the fitted models.

We generated two different models to evaluate pupal and female adult abundance using a generalized linear mixed model (GLMM) and generalized additive mixed model (GAMM) for count data (Table 2). We employed mixed models due to the potential lack of spatial independence in our data, with random intercepts for household and surveyor to account for repeated measurements and observer variability. Model coefficients were exponentiated and interpreted as incidence rate ratios [36,37]. Zero-inflated models were evaluated to determine whether the data-generating process included structural zeros beyond those expected from the count distribution.

Container productivity analysis. The association between *Ae. aegypti* pupal abundance and water storage practices was evaluated fitting a GLMM with a Poisson distribution and log link function. Fixed effects included Community (2

**Table 1. Entomological indices of *Ae. aegypti* and their definitions.**

| Stegomyia indices | Definition |
|---|---|
| House index (HI) | (# houses with immature *Ae. aegypti*/total houses inspected) * 100 |
| Container index (CI) | (# containers with immature *Ae. aegypti*/total containers per house) *100 |
| Breteau index (BI) | (# of containers with immature *Ae. aegypti*/total houses inspected) *100 |
| **Pupal indices** | |
| Pupae per house index (PHI) | # of *Ae. aegypti* pupae/inspected houses |
| Pupae per container index (PCI) | # of *Ae. aegypti* pupae/inspected containers |
| Pupae per person index (PPI) | # of *Ae. aegypti* pupae/total population of inspected houses |
| **Adult indices** | |
| Adult index (AI) | (# of houses positive for adult *Ae. aegypti*/total houses inspected) *100 |
| Adults per person (AP) | # of adult *Ae. aegypti*/total population of inspected houses |
| Adults per house (AH) | # of adult *Ae. aegypti*/total houses inspected |
| Females per person (FP) | # of adult females *Ae. aegypti*/total population of inspected houses |

**Table 2. Generalized linear and generalized additive mixed models, fixed and random effect structure, Akaike information criterion (AIC), and Bayesian information criterion (BIC).**

| Type | Smooth | Fixed | Random | AIC – BIC |
|---|---|---|---|---|
| GLMM: Pupae | | Community + season + year + water intermittency + barrels + cement wash basins + tires + buckets | Household + surveyor | 9184 – 9251 |
| GLMM: Pupae | | Community + season + year + covered barrel + partial covered barrel + uncovered barrel | Household + surveyor | 9246 – 9296 |
| GLMM: Female | | Total pupae + community + season + year + repellent + bed nets + water intermittency + trash + total water-holding containers + temperature + humidity | Household + surveyor | 2855 – 2956 |
| GAMM: Female | Total pupae | Community + season + year + repellent + bed nets + water intermittency + trash + total water-holding containers + temperature + humidity | Household + surveyor | 2857 – 2963 |

levels = Urban and Rural), Season (2 levels = Dry and Rainy), Year (2 levels = 2022 and 2023), water intermittency issues (2 levels = Yes and No), and the number of water-holding containers by type, including barrels, cement washbasins, tires, and buckets. These variables were included to evaluate how container presence and water availability contributed to pupal production at the household level. Following the previous modeling approach, we explored how covered, partially covered, and uncovered barrel management practices modulated pupal abundance.

Female adult mosquito abundance. Adult female *Ae. aegypti* abundance was analyzed using generalized linear mixed models (GLMM) and generalized additive mixed models (GAMM) for count data with a Poisson error distribution and log link function. Fixed effects included Community (2 levels = Urban and Rural), Season (2 levels = Dry and Rainy), Year (2 levels = 2022 and 2023), use of repellents (2 levels = Yes and No), use of bednets (2 levels = Yes and No), water intermittency issues (2 levels = Yes and No), trash management (5 levels = Burn, Bury, Collected, Throw in channel, Throw in empty lot), total number of water-holding containers, and environmental conditions such as household temperature and humidity. To account for variation in household density, we additionally evaluated a GLMM of female mosquitoes per person by following the previous modeling approach using household density as an offset term, allowing the model to estimate the expected number of female mosquitoes relative to the number of residents within each household.

To evaluate the association between immature mosquito abundance and adult populations, the total number of pupae collected per household was included as a predictor. Two model structures were compared. In the GLMM, pupal abundance was included as a linear fixed effect. In the GAMM, pupal abundance was modeled using a spline function to allow for potential nonlinear relationships between immature stages and adult mosquito abundance. We used the 'gamm4' packages [38] for this comparison.

For a detailed step-by-step procedure, see the "Availability of data and material" section for the link to our code repository, which includes further explanation regarding model selection. Data heteroscedasticity was evaluated by plotting the residuals as a function of predicted values for the distribution models. Model fit was assessed using Akaike Information Criterion (AIC) [39], Bayesian Information Criterion (BIC), residual diagnostics, and dispersion statistics [40,41]. All models were generated with R 4.4.1 (R Core Team, Vienna, Austria) using the 'glmmTMB' [42] and 'gamm4' packages [38]. All study data collected underwent a double-entry procedure for quality assurance.

## Results

### Community demographic structure

In each of the four surveys (two in the dry season and two in the rainy season), we inspected 500 houses (250 urban and 250 rural) for a total of 2,000 inspections. Overall, we observed similar patterns for key demographics between urban and rural communities. For instance, urban communities had a mean number of people per house of 5.1 (SE = 0.13) to 5.3 (SE = 0.17) compared to rural with 4.4 (SE = 0.12) to 4.7 (SE = 0.13), female respondents were predominant in both

communities (urban: 89% female; rural: 82% female), and similar structures were observed for age and education levels (S1 Table). Regarding public utilities, rural communities relied more on burning waste (48% vs 5%), had lower access to municipal waste collection services (32% vs 87%), faced more interruptions in water supply (51% vs 35%), and lacked formal electricity meters connected to the power grid (48% vs 67%) compared to urban communities. In rural communities, we documented 1,106 buildings/km$^2$ and in urban communities, 4,940 buildings/km$^2$.

**Container productivity**

We inspected a total of 6,727 containers, with many undergoing multiple inspections. A total of 41,254 immature *Ae. aegypti* (larvae: 37,255 and pupae: 3,999) were collected (Tables 3 and S2). Overall, rural communities produced most immature forms, accounting for 70% of larvae (n = 26,046) and 74% of pupae (n = 2,962). We observed 60% of all containers in rural communities. Consistently, we found that a small subset of houses contributed disproportionately to overall immature mosquito production. In the dry season of 2022, only 1.6% (8/500) of houses produced 67.2% (3,268/4,862) of all larvae and pupae. Similar patterns were observed in the 2022 rainy season (6.2% of houses producing 63.8% [7,945/12,455] of immatures), the 2023 dry season (3.2% of houses producing 52.8% [3,624/6,861]), and the 2023 rainy season (11.2% of houses producing 68.6% [11,710/17,076]). These highly productive houses typically harbored ≥100 larvae and at least one pupa. Containers were not individually marked, preventing us from quantifying replacements over time.

Houses in rural communities consistently had more containers than those in urban communities, regardless of season or year. During the dry seasons of 2022 and 2023, pupal positivity was relatively low in both urban and rural communities (urban: 2.8–6.1%, rural: 2.7–6.5%), though rural houses averaged a higher number of containers per house (2022: 3.8 [SE = 0.02]; 2023: 4.2 [SE = 0.04]) (Table 3). Unsurprisingly, during the rainy seasons, pupal positivity increased, to

**Table 3. Entomological collections of immature *Ae. aegypti* in urban and rural settings of Managua, Nicaragua, during dry and rainy seasons of 2022 and 2023.**

| Study site | Season | Year | Houses with cont. | # Cont. inspected | # Cont. per house (SE) | # Cont. + pupae | Larvae | Pupae |
|---|---|---|---|---|---|---|---|---|
| **Rural** | Dry | 2022 | 250 | 943 | 3.8 (0.29) | 25 (2.7%) | 3,303 (12.7%) | 270 (9.1%) |
| | Rainy | 2022 | 250 | 785 | 3.1 (0.13) | 108 (13.8%) | 7,684 (29.5%) | 966 (32.6%) |
| | Dry | 2023 | 249 | 1,047 | 4.2 (0.29) | 68 (6.5%) | 4,438 (17.0%) | 466 (15.7%) |
| | Rainy | 2023 | 247 | 1,294 | 5.2 (0.32) | 200 (15.5%) | 10,621 (40.8%) | 1,260 (42.5%) |
| | | Total | | 4,069 | | 401 | 26,046 (100.0%) | 2,962 (100.0%) |
| **Urban** | Dry | 2022 | 250 | 604 | 2.4 (0.16) | 17 (2.8%) | 1,232 (11.0%) | 57 (5.5%) |
| | Rainy | 2022 | 250 | 608 | 2.4 (0.08) | 45 (7.4%) | 3,577 (31.9%) | 228 (22.0%) |
| | Dry | 2023 | 250 | 659 | 2.6 (0.14) | 40 (6.1%) | 1,708 (15.2%) | 249 (24.0%) |
| | Rainy | 2023 | 249 | 787 | 3.1 (0.16) | 85 (10.8%) | 4,692 (41.9%) | 503 (48.5%) |
| | | Total | | 2,658 | | 187 | 11,209 (100.0%) | 1,037 (100.0%) |

Cont.= Containers; SE= standard error.

7.4–10.8% in urban and 13.8–15.5% in rural houses. Overall, rural houses not only stored more containers but also had a higher proportion of pupal positive containers during the rainy season.

The GLMM showed several factors significantly associated with *Ae. aegypti* pupal abundance (Table 4). Rural households had 3.34 times higher pupal counts compared with urban households (95% CI: 2.17–5.15, p<0.001). Households that reported water intermittency issues had 30% higher pupal counts (IRR=1.30, 95% CI: 1.15–1.47, p<0.001). Importantly, the type of water-holding container had distinct associations with pupal abundance. The containers most productive for pupae, regardless of year and season, were barrels (55–79%), followed by buckets (0.4–32%) and, to a lesser extent, cement laundry sinks (0–9%) (Fig 2). Barrels showed the highest association with pupal abundance; each additional barrel within a household increased pupal counts by 40% (IRR=1.40, 95% CI: 1.34–1.46, p<0.001). This pattern was also

**Table 4. Fixed-effect estimates from a generalized linear mixed model evaluating *Ae. aegypti* pupal abundance and water management.**

| Variable | Exp (β) | β (SE) | 95% CI | Z value | p-value |
|---|---|---|---|---|---|
| Season (Rainy) | 2.93 | 1.08 (0.16) | 2.64 – 3.25 | 20.26 | <0.001 |
| Year (2023) | 1.67 | 0.52 (0.09) | 1.50 – 1.86 | 9.25 | <0.001 |
| Community (Rural) | 3.34 | 1.21 (0.73) | 2.17 – 5.15 | 5.47 | <0.001 |
| Water intermittency | 1.30 | 0.26 (0.08) | 1.15 – 1.47 | 4.3 | <0.001 |
| Barrels | 1.40 | 0.34 (0.03) | 1.34 – 1.46 | 15.57 | <0.001 |
| Concrete washbasins | 0.74 | -0.30 (0.05) | 0.65 – 0.84 | -4.5 | <0.001 |
| Tires | 1.15 | 0.14 (0.02) | 1.12 – 1.19 | 9.2 | <0.001 |
| Buckets | 1.00 | 0.01 (0.01) | 0.98 – 1.03 | 0.31 | 0.756 |

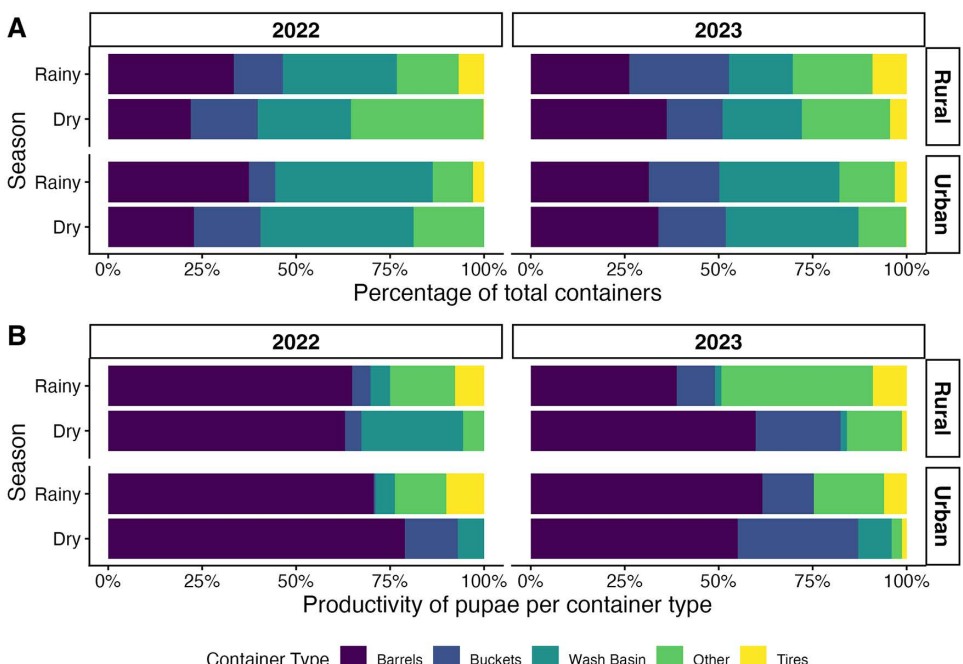

**Fig 2. *Ae. aegypti* pupae collected in rural and urban communities of Managua, Nicaragua, during the rainy and dry seasons of 2022 and 2023. A)** Percentage of total containers found in the communities. **B)** Productivity of pupae per container. Green=other containers (e.g., jugs, animal water dishes, flowerpots, pieces of plastic, scrap metal, etc.).

observed for tires, with a 15% increase per tire (IRR = 1.15, 95% CI: 1.12–1.19, p < 0.001). In contrast, concrete washbasins were associated with a 26% decrease in pupal counts (IRR = 0.74, 95% CI: 0.65–0.84, p < 0.001). We did not observe a significant association for buckets.

We also evaluated how *Ae. aegypti* pupal abundance is modulated by management of barrels in a household. We observed that partially covered barrels were significantly associated with increased pupal counts, with each additional partially covered barrel increasing pupal abundance by approximately 30% (IRR = 1.29, 95% CI: 1.20–1.40, p < 0.001). The strongest association was observed for uncovered barrels, where each additional uncovered barrel increased pupal abundance by approximately 73% (IRR = 1.73, 95% CI: 1.63–1.82, p < 0.001) (Table 5). No association was observed for covered barrels.

### Entomological indices of *Ae. aegypti*

***Stegomyia* and pupal indices.** Across nearly all surveys, both traditional *Stegomyia* indices and pupal indices were higher in rural communities compared to urban ones (Fig 3 and S4-S12 Tables). The only exception was the 2022 dry season, when values were similar between settings. As previously observed, *Stegomyia* indices were highest during the rainy season; in rural communities, the house index (HI) (Fig 3A) ranged from 59.2 to 58.8, the container index (CI) (Fig 3B) ranged from 36.2 to 32.3, and the Breteau index (BI) (Fig 3C) ranged from 113.6 to 167.2 in 2022 and 2023, respectively. Urban indices were markedly lower during the same periods (HI: 42.8–41.2; CI: 28.1–23.9; BI: 68.4–75.2). A similar pattern was observed for pupal indices. In rural communities, the pupal house index (PHI) (Fig 3D) ranged from 3.86 to 5.04, the pupal container index (PCI) (Fig 3E) from 1.23 to 0.97, and the pupal per person index (PPI) (Fig 3F) from 0.82 to 1.12 in 2022 and 2023, respectively. Again, urban values were substantially lower (PHI: 0.91–2.01; PCI: 0.38–0.64; PPI: 0.17–0.38). Despite the dominance of a few highly productive houses, immature *Ae. aegypti* were detected throughout both urban and rural communities, underscoring the widespread nature of infestation. Together, these findings highlight the major contribution of rural communities to immature *Ae. aegypti* abundance and showcase the heightened transmission potential in these settings, particularly during the rainy season.

**Adult indices.** We collected a total of 1,468 adult *Ae. aegypti*. Both females and males were more abundant in rural communities, which accounted for 62% of all female captures and 59% of all male captures. Seasonal patterns were consistent across settings, with the majority of adults collected during the rainy season: > 70% of females and males in both urban (females: 77% [n = 211]; males: 74% [n = 228]) and rural (females: 73% [n = 321]; males: 72% [n = 318]) communities (S3 Table). The adult index (AI) showed small differences between rural and urban communities during the dry seasons of 2022 (Rural: 18.8% vs. Urban: 14.0%) and 2023 (Rural: 32.0% vs. Urban: 25.2%) (Fig 4A). In the rainy season of 2022, however, rural households were substantially more likely to be positive (Rural: 51.2% vs. Urban: 34.8%). By contrast, in the rainy season of 2023, positivity was similar in both settings (Rural: 53.2% vs. Urban: 50.8%).

**Table 5. Fixed-effect estimates from a generalized linear mixed model evaluating the association between household barrel management and *Ae. aegypti* pupal abundance.**

| Variable | Exp (β) | β (SE) | 95% CI | Z value | p-value |
|---|---|---|---|---|---|
| Season (Rainy) | 2.65 | 0.98 (0.14) | 2.40 – 2.95 | 18.65 | <0.001 |
| Year (2023) | 2.03 | 0.71 (0.11) | 1.83 – 2.26 | 11.5 | <0.001 |
| Community (Rural) | 3.74 | 1.32 (0.82) | 2.44 – 5.75 | 6.03 | <0.001 |
| Covered barrels | 0.97 | -0.02 (0.03) | 0.92 – 1.05 | -0.61 | 0.542 |
| Partially covered barrels | 1.29 | 0.26 (0.04) | 1.20 – 1.40 | 6.64 | <0.001 |
| Uncovered barrels | 1.73 | 0.55 (0.03) | 1.63 – 1.82 | 19.58 | <0.001 |

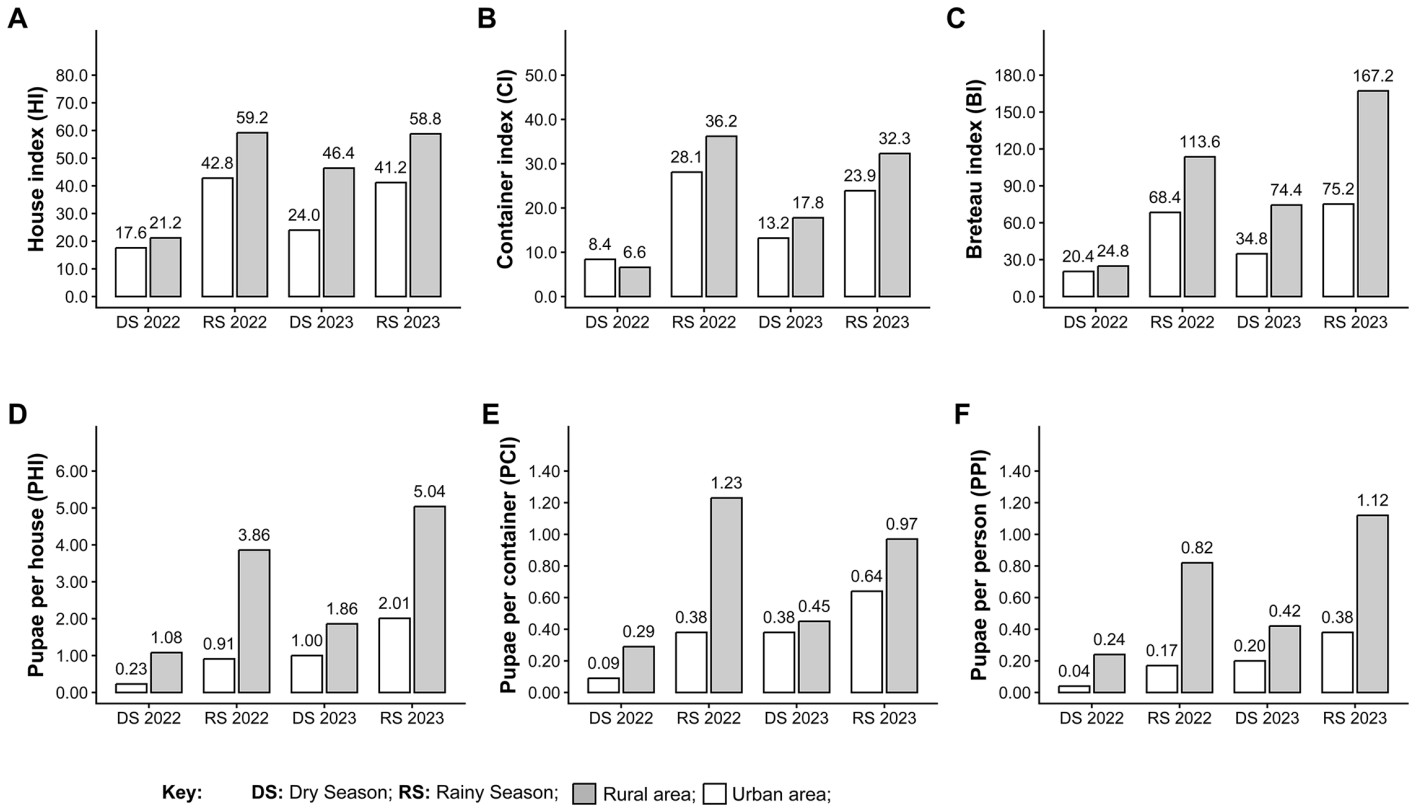

**Fig 3. Entomological indices measured in rural and urban communities of District III of Managua, Nicaragua. A**) House index (HI). **B**) Container index (CI). **C**) Breteau index (BI). **D**) Pupae per house index (PHI). **E**) Pupae per containers index (PCI). **F**) Pupae per person index (PPI).

For the number of adults per house (AH) (Fig 4B), we observed that rural households contained more adults than urban households. This was observed across all seasons for both 2022 (dry season [Rural: 0.32 vs. Urban: 0.16] and rainy season [Rural: 1.20 vs. Urban: 0.58]) and 2023 (dry season [Rural: 0.66 vs. Urban: 0.42] and rainy season [Rural: 1.36 vs. Urban: 1.18]). As before, adults per person (AP) (Fig 4C), was higher in rural communities compared to urban ones across all seasons for both 2022 (dry season [Rural: 0.07 vs. Urban: 0.03] and rainy season [Rural: 0.26 vs. Urban: 0.11]) and 2023 (dry season [Rural: 0.15 vs. Urban: 0.08] and rainy season [Rural: 0.30 vs. Urban: 0.22]). The female per person (FP) (Fig 4D) values showed the same trend, with rural communities having higher FP values than urban ones across all seasons for both 2022 (dry [Rural: 0.03 vs. Urban: 0.01] and rainy season [Rural: 0.12 vs. Urban: 0.04]) and 2023 (dry [Rural: 0.08 vs. Urban: 0.04] and rainy season [Rural: 0.16 vs. Urban: 0.12]).

### Female adult *Ae. aegypti* abundance

We next evaluated factors associated with female adult *Ae. aegypti* abundance using generalized linear and generalized additive mixed models with a Poisson distribution and log link function. The best fit model after backward elimination (AIC = 2846.0, BIC = 2901.9) included a smoothed term for pupal abundance and main effects for season, year, community type, water-holding containers, and average humidity. Pupal abundance showed a strong non-linear association with adult mosquito counts (edf = 3.86, $\chi^2$ = 103.9, *p* < 0.001), indicating that increases in pupae were associated with higher adult abundance, in a non-linear way (Fig 5A). We observed that the relationship between pupal counts and female abundance showed a threshold-like pattern: an initial increase in adult females with 1–18 pupae, a plateau between ~19–41 pupae,

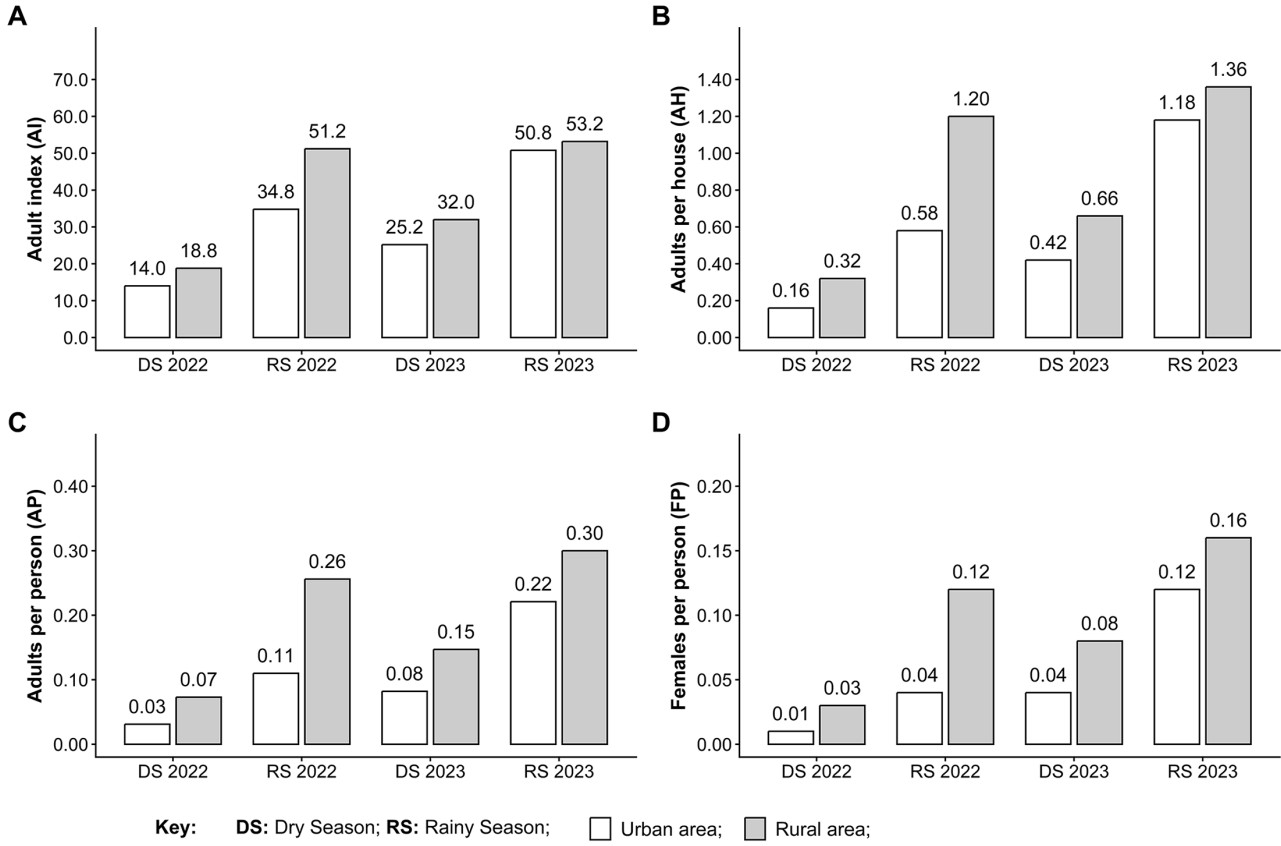

**Fig 4. Adult *Aedes aegypti* house indices for rural and urban communities of District III of Managua, Nicaragua. A**) Houses with adult *Ae. aegypti* (AI). **B**) Adult *Ae. aegypti* per house (AH). **C**) Adult Ae. aegypti per person (AP). **D**) Female Ae. aegypti per person (FP).

and a sharp rise beyond 42 pupae. Predicted female counts increased from 0.14 (95% CI = 0.10–0.20) at 1 pupa to 0.28 (95% CI = 0.18–0.44) at 13 pupae and remained stable up to 0.42 (95% CI = 0.23–0.77) at 50 pupae -- and then rose to 2.22 females (95% CI = 0.82–6.04) at 84 pupae (Fig 5B). This suggests the presence of pupal thresholds that may drive sharp increases in adult abundance once exceeded.

Several environmental and temporal variables were also significant predictors of female adult abundance. As consistently observed, community type was also associated with female adult mosquito abundance. Rural communities had 39% higher female adult counts compared with urban communities (IRR = 1.39, 95% CI = 1.13–1.72, $p < 0.001$). Female counts were 2.73 times (IRR = 2.73, 95% CI = 2.04–3.65, $p < 0.001$) higher during the rainy season compared with the dry season. Similarly, female abundance increased 2.04 times in 2023 relative to 2022 (IRR = 2.04, 95% CI = 1.64–2.54, $p < 0.001$) (Table 6). Other fixed effects, such as household water storage and humidity, show limited impact on female abundance. Each additional water-holding container was associated with a small 4% increase in female counts (IRR = 1.04, 95% CI = 1.02–1.06, $p < 0.001$). In contrast, higher average humidity was associated with a small 1% decrease in adult abundance (IRR = 0.99, 95% CI = 0.97–0.998, $p = 0.02$). We did not observe significance for household density, the use of mosquito repellent, bed nets, water intermittency, trash management, or temperature. Overall, our model indicates that female *Ae. aegypti* abundance is strongly associated with pupal production, seasonal conditions, and community context (rural), while household water availability and humidity also contribute to observed variation in a smaller portion. Lastly,

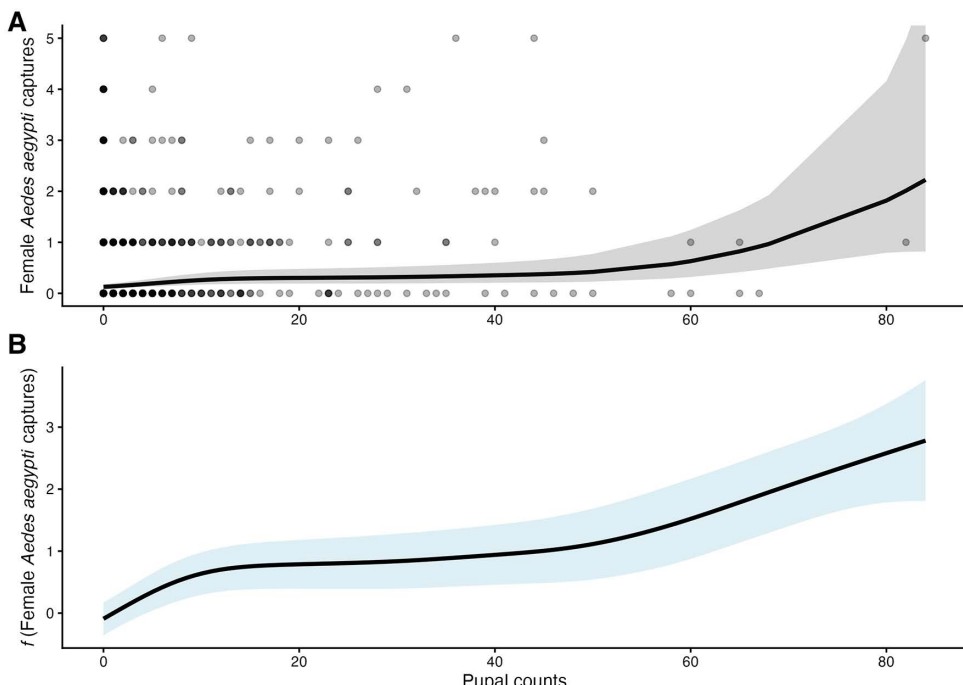

**Fig 5. Generalized Additive Mixed Model (GAMM) of pupal counts and female *Ae. aegypti* captured in District III of Managua, Nicaragua. A)** Predicted adult female counts across pupal gradients with 95% CI and observed data overlay. **B)** Smoothed GAMM function.

**Table 6. Fixed-effect estimates from a generalized additive mixed model evaluating factors associated with female adult *Ae. aegypti* abundance.**

| Variable | Exp (β) | β (SE) | 95% CI | Z value | p-value |
|---|---|---|---|---|---|
| Season (Rainy) | 2.73 | 1.00 (0.149) | 2.04 – 3.65 | 6.74 | <0.001 |
| Year (2023) | 2.04 | 0.72 (0.111) | 1.64 – 2.54 | 6.42 | <0.001 |
| Community (Rural) | 1.39 | 0.32 (0.105) | 1.13 – 1.72 | 3.11 | 0.002 |
| Water-holding containers | 1.04 | 0.04 (0.010) | 1.02 – 1.06 | 3.90 | <0.001 |
| Average humidity | 0.99 | -0.02 (0.006) | 0.97 – 0.998 | -2.31 | 0.021 |

after accounting for household population size using an offset term, female *Ae. aegypti* abundance per household resident was significantly higher in rural households (IRR = 1.67, 95% CI: 1.33–2.09) compared to urban ones (S14 Table).

## Discussion

*Ae. aegypti* remains a major public health threat in tropical regions and an emerging threat globally, driven in part by its ecological range expansion as a consequence of climate change [1,4,43]. With increasing dengue epidemics and billions of people at risk of arbovirus infection, a fine-scale understanding of *Ae. aegypti* distribution is urgently needed, particularly in communities traditionally overlooked in mosquito surveillance, such as rural areas [44,45]. Our study in District III of Managua, Nicaragua, demonstrates that rural communities consistently had higher *Ae. aegypti* abundance than urban ones across multiple entomological indicators. Importantly, while mosquito presence was widespread across both settings, mosquito abundance was consistently higher in rural households, highlighting the importance of expanding mosquito surveillance and vector control activities beyond traditional urban settings of Managua and other similar locations. These

findings support a growing body of evidence that suggests *Ae. aegypti* populations are shaped by local ecological and social factors rather than by urbanicity alone [46].

Strong seasonal effects were observed, with pupal counts peaking during the rainy season, reflecting greater water availability and increased suitability of larval development habitats. Rural households, with greater reliance on water storage, exhibited higher pupal productivity per household. Barrels were consistently identified as the most productive containers in both urban and rural communities, which is consistent with observations from Mexico and Argentina, where larger containers also dominate mosquito production [47,48]. Similar results in terms of rural *Ae. aegypti* infestation were observed in a longitudinal study in Kenya, with higher pupal density per container in Western rural sites [49]. However, this contrasts with results observed in Colombia [21] and Sri Lanka [50], where urban communities showed higher pupal densities and higher container indices compared to rural areas. We also consistently observed that *Stegomyia* and pupal indices in rural communities were higher than those in urban areas. Similar findings have been observed in Zanzibar [51] but contrast with reports from Cambodia [52] and other parts of Latin America, where urban sites showed higher infestation [21,53,54]. These differences highlight how ecological and social conditions in combination with infrastructure shape mosquito ecology across different geographic regions.

A key finding of our study was that relatively few households contributed disproportionally to mosquito production, consistent with hotspot dynamics reported elsewhere [55]. Across seasons and years, a small fraction of houses accounted for more than half of all immature mosquitoes. This pattern of "key sites" has also been reported in other settings showing that these locations can drive mosquito population dynamics [55]. From a public health perspective, these findings reinforce the value of targeted interventions aimed at highly productive households and containers, as it has been previously successfully implemented in other areas of Managua [12,13]. However, adult mosquito captures were more evenly distributed across the landscape, likely reflecting dispersal from productive households, key sites, and containers, as has been documented in mark–release–recapture studies of *Ae. aegypti* [56]. This suggests that while hotspot elimination may reduce larval productivity, area-wide and community-based approaches remain essential for controlling mobile adult populations.

Our mixed-model analyses provided additional insight into the ecological drivers of mosquito abundance. Barrel management was strongly associated with pupal production, particularly when containers were partially covered or left uncovered. Each additional uncovered barrel substantially increased pupal counts, suggesting that water storage practices remain one of the most important drivers of mosquito productivity in these communities. These results reinforce previous work in Nicaragua demonstrating that household water storage and intermittent water supply contribute significantly to vector immature mosquito development [15]. In addition to container-level effects, our modeling framework revealed nonlinear relationships between pupal abundance and adult female mosquitoes. Using generalized additive mixed models, we identified threshold-like dynamics in which adult abundance initially increased with pupal counts, stabilized at intermediate levels, and rose sharply once pupal numbers exceeded a critical range. Such nonlinear relationships have been suggested previously in experimental mosquito populations where density-dependent processes influence survival and adult emergence [57–60]. These dynamics suggest that containers surpassing certain pupal densities may disproportionately contribute to adult mosquito populations and therefore represent particularly important targets for vector control. Notably, our pupal indices remained within transmission-relevant thresholds across both dry and rainy seasons [61].

Additionally, we observed that environmental and temporal factors also influenced female adult abundance. Community type played a key role as well, with rural areas showing consistent higher mosquito densities even after accounting for environmental variables. We were able to consistently observe that both total female counts and female per person rates were significantly higher in rural communities compared to urban ones, with up to 40–67% increments in rural communities. Additionally, while humidity and total number of water-holding containers showed a statistical association, their biological effects were relatively small compared to seasonal and pupal drivers. Female *Ae. aegypti* were more abundant during

the rainy season and in 2023, reflecting broader climatic conditions favorable to mosquito development. Overall, these results show that mosquito abundance in Managua is driven by a mixture of seasonal dynamics, water management practices, and localized ecological conditions. Furthermore, deconvoluting the dynamics driving vector abundance and disease transmission has been a long-standing endeavor of our group, where we examine the integrated role of landscape, biological, and social determinants, referred to as the SCAPES framework.

A limitation of our study was that the collection only occurred at specific time points; thus, the lack of longer-term temporal data prevents a deeper analysis of the ecological dynamics of these urban and rural communities. However, we compensate for this drawback with a robust sample size and a thorough entomological sampling of the houses surveyed. We believe that even though our temporal collection events are restricted to seasons, they provide enough evidence to highlight the importance of rural communities in sustaining *Ae. aegypti* populations and the potential to drive arboviral disease transmission in such areas. However, during this time-period, our mosquito-based arbovirus surveillance only yielded positive pools in the urban communities [62], likely due to greater viral circulation and population density. We also acknowledge that this study did not evaluate non-residential larval development sites (i.e., schools, cemeteries, churches, tire shops, etc.), which can play a role in sustaining mosquito populations in the area. We are currently evaluating the impact of non-residential sites as drivers of population dynamics in District III. Of note, in the 2023 rainy season, we recorded for the first time adult *Ae. albopictus* in 46 households (urban: 13; rural: 33) of District III, showcasing the need for further surveillance efforts to evaluate the impact that *Ae. albopictus* might have in the ecology of *Ae. aegypti* and the transmission of arboviruses.

Importantly, current trends indicate that global climatic conditions are increasingly favorable for the expansion of *Ae. aegypti*, alongside growing environmental suitability for arboviral diseases such as dengue [63]. As a result, regions where *Ae. aegypti* is already established but where dengue transmission remains limited or absent [64] may face the highest epidemic risk. This is particularly concerning for rural or remote communities, where populations are likely to be immunologically naïve and public health infrastructure may be limited.

## Conclusion

Overall, our findings showcase that rural communities had consistent higher abundance in most entomological endpoints for *Ae. aegypti* across seasons and years. Our results challenge the long-standing perception of *Ae. aegypti* as primarily an urban mosquito and demonstrates that rural communities can sustain substantial mosquito populations capable of supporting arbovirus transmission. Further, these results underscore the importance of moving beyond a simple urban-rural dichotomy toward a more nuanced evaluation of landscapes, as finer-scale ecological and social heterogeneity likely plays a critical role in shaping mosquito abundance. Incorporating such complexities into surveillance frameworks will be essential for developing targeted interventions and strengthening public health strategies against arbovirus transmission.

## Supporting information

**S1 Table. Demographic and household information and educational level of urban and rural households in District III of Managua, Nicaragua.**
(DOCX)

**S2 Table. Entomological collections of *Ae. aegypti* larvae in urban and rural settings of Managua, Nicaragua, during dry and rainy seasons of 2022 and 2023.**
(DOCX)

**S3 Table. Entomological collections of adult *Ae. aegypti* in urban and rural settings of Managua, Nicaragua, during dry and rainy seasons of 2022 and 2023.**
(DOCX)

**S4 Table. House index (HI).**
(DOCX)

**S5 Table. Container index (CI).**
(DOCX)

**S6 Table. Breteau index (BI).**
(DOCX)

**S7 Table. Pupae per houses index (PHI).**
(DOCX)

**S8 Table. Pupae per containers index (PCI).**
(DOCX)

**S9 Table. Pupae per persons index (PPI).**
(DOCX)

**S10 Table. Adult index (AI).**
(DOCX)

**S11 Table. Adults per house (AH).**
(DOCX)

**S12 Table. Adults per person (AP).**
(DOCX)

**S13 Table. Females per person (FP).**
(DOCX)

**S14 Table. Fixed-effect estimates from a generalized linear mixed model evaluating factors associated with female per person rates of *Aedes aegypti* abundance.**
(DOCX)

## Acknowledgments

We would like to thank all the members of our A2CARES study team in Nicaragua, in particular Meyling Escobar Cárcamo and Carlos Santos Acevedo, who were involved in the collection of entomological samples and laboratory speciation procedures; Everts Morales Reyes for the support of customized informatic systems; Jorge Ruiz Salinas for support in database management; and Juan Carlos Mercado for support with material and logistics. We thank the residents of District III who allowed entomological collections in and around their homes. Additionally, we are grateful for valuable collaboration with community leaders and overall support from local health authorities.

## Author contributions

**Conceptualization:** Harold Suazo-Laguna, Eva Harris, Josefina Coloma.

**Data curation:** Harold Suazo-Laguna, Jose G. Juarez.

**Formal analysis:** Harold Suazo-Laguna, Jose G. Juarez.

**Funding acquisition:** Josefina Coloma.

**Investigation:** Harold Suazo-Laguna, Jacqueline Mojica-Díaz, María M. Lopez.

**Methodology:** Harold Suazo-Laguna, Jacqueline Mojica-Díaz, María M. Lopez, Eva Harris, Josefina Coloma, Jose G. Juarez.

**Project administration:** Jacqueline Mojica-Díaz, Angel Balmaseda, Eva Harris, Jose G. Juarez.

**Resources:** Angel Balmaseda, Eva Harris, Josefina Coloma.

**Supervision:** Harold Suazo-Laguna, Jacqueline Mojica-Díaz, María M. Lopez, Angel Balmaseda, Jose G. Juarez.

**Validation:** María M. Lopez, Jose G. Juarez.

**Visualization:** Harold Suazo-Laguna, Jose G. Juarez.

**Writing – original draft:** Harold Suazo-Laguna, Eva Harris, Jose G. Juarez.

**Writing – review & editing:** Jacqueline Mojica-Díaz, María M. Lopez, Angel Balmaseda, Josefina Coloma.

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
