## [Decision Letter · Decision Letter 0]

3 Sep 2025

PNTD-D-25-00902

Abundance of the vector Aedes aegypti in urban and rural areas in Managua, Nicaragua

Dear Dr. Juarez,

Thank you for submitting your manuscript to PLOS Neglected Tropical Diseases. After careful consideration, we feel that it has merit but does not fully meet PLOS Neglected Tropical Diseases's publication criteria as it currently stands. Therefore, we invite you to submit a revised version of the manuscript that addresses the points raised during the review process.

Please submit your revised manuscript within 30 days Nov 02 2025 11:59PM. If you will need more time than this to complete your revisions, please reply to this message or contact the journal office at plosntds@plos.org. Please include the following items when submitting your revised manuscript:

* A rebuttal letter that responds to each point raised by the editor and reviewer(s). You should upload this letter as a separate file labeled 'Response to Reviewers'. This file does not need to include responses to any formatting updates and technical items listed in the 'Journal Requirements' section below.'. This file does not need to include responses to any formatting updates and technical items listed in the 'Journal Requirements' section below.

* A marked-up copy of your manuscript that highlights changes made to the original version. You should upload this as a separate file labeled 'Revised Manuscript with Track Changes'.'.

* An unmarked version of your revised paper without tracked changes. You should upload this as a separate file labeled 'Manuscript'.'.

We look forward to receiving your revised manuscript.

Kind regards,

Geoffrey M. Attardo

Academic Editor

Audrey Lenhart

Section Editor

Shaden Kamhawi

co-Editor-in-Chief

Paul Brindley

co-Editor-in-Chief

**Additional Editor Comments:**

Dear Dr. Suazo-Laguna and colleagues,

Thank you for your submission to PLOS Neglected Tropical Diseases. Your manuscript, investigating Aedes aegypti densities in rural versus urban settings, addresses an understudied topic with direct relevance to public health in the region and beyond. Both reviewers recognize the importance of your findings.

After careful consideration of the reviews, I am pleased to inform you that we have made a decision of “minor revision” on your manuscript. The key results are sound and relevant. However, the reviewers had some suggestions that, if addressed, will significantly strengthen the clarity, contextualization, and impact of your paper.

Below is a summary of issues raised by the reviewers that should be addressed:

Both reviewers emphasized the need to clearly define and contextualize the urban and rural classifications. Please provide a more detailed description of the physical and infrastructure differences between the settings. A table contrasting features (e.g., housing density, drainage, garbage collection, household spacing, presence of screens or gutters, vegetation, etc.) and/or representative photographs would help clarify.

Reviewer 1 would like more detail on the spatial distribution of sampled houses. Please consider including a map that indicates all houses surveyed (not just those with adult mosquito detections) and comment on the proximity of houses relative to Aedes aegypti dispersal ranges.

Reviewer 2 suggested enhancing the clarity and impact of your findings related to container productivity and Stegomyia indices. Please consider including per-house estimates in the container productivity table and revisiting the description and implications of the indices reported in Figure 3 (including adding index names to the figure itself)

Reviewer 2 found the manuscript overly verbose at times. Please revise for conciseness and stronger narrative focus, particularly in the results and discussion sections. Highlight the key findings and emphasize the implications for vectorial capacity.

Please ensure clarity in your methods regarding:

• Which larval stages were counted (e.g., only L4 and pupae?)

• The absence (or presence) of Ae. albopictus

• Aspirator model (change to “Prokopack” as per R2)

• Index definitions—adding a table for clarity is encouraged

Reviewer 1 encouraged the sharing of per-household data (with identifying details removed) and the inclusion of spatial data if possible. We strongly encourage making this information available in supplementary materials or in an open repository to enhance the value of your work for reuse and meta-analysis.

Please consider addressing the following:

• Seasonal patterns and inter-annual variation in mosquito density

• Potential roles of vegetation or ornamental plants as breeding habitats

• Non-residential breeding sites and their role in sustaining Ae. aegypti populations

Please revise your manuscript to address the above points and provide a detailed point-by-point response to each reviewer's comment. Be sure to indicate where changes have been made in the manuscript text.

We look forward to receiving your revised manuscript. Please do not hesitate to contact me if you have any questions or need clarification on the requested revisions.

**Journal Requirements:**

1) Please provide an Author Summary. This should appear in your manuscript between the Abstract (if applicable) and the Introduction, and should be 150-200 words long. The aim should be to make your findings accessible to a wide audience that includes both scientists and non-scientists. Sample summaries can be found on our website under Submission Guidelines:

- ® on page: 7.

Potential Copyright Issues:

i) Figure 1. Please (a) provide a direct link to the base layer of the map (i.e., the country or region border shape) and ensure this is also included in the figure legend; and (b) provide a link to the terms of use / license information for the base layer image or shapefile. We cannot publish proprietary or copyrighted maps (e.g. Google Maps, Mapquest) and the terms of use for your map base layer must be compatible with our CC BY 4.0 license.

7) Kindly revise your competing statement to align with the journal's style guidelines: 'The authors declare that there are no competing interests.'

**Reviewers' comments:**

Reviewer's Responses to Questions

**Key Review Criteria Required for Acceptance?**

**Methods**

-Are the objectives of the study clearly articulated with a clear testable hypothesis stated?

-Is the study design appropriate to address the stated objectives?

-Is the population clearly described and appropriate for the hypothesis being tested?

-Is the sample size sufficient to ensure adequate power to address the hypothesis being tested?

-Were correct statistical analysis used to support conclusions?

-Are there concerns about ethical or regulatory requirements being met?

Reviewer #1: The survey methodology was rigorous and well planned but limited in scale (only 2 years and a total of 4 surveys). Statistical analysis appears appropriate, and I have no ethical concerns. No meteorological analysis was attempted, although I accept that two surveys were conducted during the dry season and two surveys during the wet season. Some description of average rainfall totals for the two seasons would be useful context.

Reviewer #2: In this paper, Suazo-Laguna et al explore the density of adults and larvae in rural vs urban residences (houses/yards). It is highly impressive with ~1000 houses and highlights that Aedes aegypti (an extremely important disease vector) numbers were higher in rural areas then the authors expected (although I note believe, it would help to better convey to readers in this Nicaragua context, what is urban and what is rural)

I think this is a good manuscript with important findings.

I provide a number of specific and more general comments below to help improve the manuscript.

Specific Comments:

Line 20: consider adding yellow fever

Line 109: is it “classification system” ? Plural “classifications? “

Line 109: I expand this in my point 1 below, but what is the definition of urban and rural? I think it would help to spell it out or picture it, etc. From a global level I would imagine what is urban and what is rural is highly subjective and varies by country.

Line 135: “the MOH also carried out” ?

Linew 135-141 : Was this at same houses? Or in the district, in general to some of the houses?

Line 185: Breteau index, Citation?

Line 217 “.R -space’ code” ?

Line 432- 433: How close were your houses? Aedes is not thought to travel very far (perhaps as low as ~100 meters in their lifetime) Does that change interpretation?

Lines 99-131/ Figure 1 : See above, here and/or Line 432-433 please consider discussing how close your collections houses are to each other/ their density. Is it possible to have a map of the collection sites showing appx locations of the collection sites ? It is unclear to me from Fig 1 all the collection sites since it only shows where adults were found - not all the sampling places.

Line 433-434 : see above comment, Aedes control can be hyper-local (at the house level) to make a difference. My concern is that what you actually have to do is find the key spots, but only area-wide management is going to locate these places.

Line 469-472: Can you share some examples of types of non-residential locations one/ you might/are consider investigating?

Line 472: Perhaps note how many occurrences (e.g. how many houses were they found? ) to give an idea of establishment of this species?

Figure 3: The figure would be enhanced for ease of reader by listing the measure (e.g. House Index (HI) in the figure itself, instead of / in addition to the figure legend.

Other questions and points:

1. Could some pictures of typical residences or breeding sites be provided in rural / urban context? Would that help? I’m thinking about Lines 223-236 where you note the demographic structure of the communities, but understanding the physical landscape differences of the structures, properties, etc would be nice. How “developed” (or not) are these. Do houses tend to have drains/ Gutters? Is trash in yard common, or not? {information on trash collection services might further contextualize this). Do houses tend to be screened or not? How close (generally) is one house to another in rural vs urban?

2. It would seem mosquito numbers went up from 2022 to 2023? Is it worth noting any thoughts on why?

3. Did you consider vegetation as larval source? Some plants can be larval sites in them self (e.g. bromeliads) Also it has been reported that irrigation etc for the purposes of maintaining decorative gardens led to increased larval density in ‘wealthier’ parts of town. Do your urban residents keep plants? Are they un-unidentified sources of larva?

**Results**

-Does the analysis presented match the analysis plan?

-Are the results clearly and completely presented?

-Are the figures (Tables, Images) of sufficient quality for clarity?

Reviewer #1: Yes overall, but the text of the article should be streamlined and reduced considerably. There is a lot of text and descriptive work without telling a story. Take home about Aedes densities being higher is rural than urban locations is clear, but not a lot more to say here.

Reviewer #2: yes

**Conclusions**

-Are the conclusions supported by the data presented?

-Are the limitations of analysis clearly described?

-Do the authors discuss how these data can be helpful to advance our understanding of the topic under study?

-Is public health relevance addressed?

Reviewer #1: Overall, yes, but the presentation should be far more concise. Found much of the text, did not have a clear point being made, and more effort to contrast the two sites are needed beyond stating that one is rural and the other urban based on National definitions. Both sites seem urban to me.

Reviewer #2: yes

**Editorial and Data Presentation Modifications?**

Reviewer #1: (No Response)

Reviewer #2: Data presentation:

I would argue that the authors should be able to share per-residence data of each measure collected. I am assuming this would not have GPS / address data. This would improve reuse.

if some sort of spatial information (such as obfuscated GPS points could be provided, that would be even better.)

**Summary and General Comments**

Reviewer #1: General Comments

1. The findings that Aedes aegypti densities are higher in rural compared to urban communities is important and few comparisons have been published. I would, however, suggest that the manuscript be streamlined to focus on this point.

2. The methodology was rigorous and analysis appropriate, but the study was rather small, covering two years and two sites only. I recognize this cannot be changed but this represents only four surveys, albeit done well. Again, this information is valuable, but this manuscript needs to emphasize differences between the rural versus urban sites, which appears to be principally during the rainy season where there are more (useless containers).

3. The issue of urban versus rural is not very convincing here, as the population in the “rural” area is a little less than half of the urban area. We need a better comparison which probably exists. Items, like housing density and infrastructure, would provide much needed context. If the comparison of the data used from satellites would be useful. The figures are very nice but don’t give a strong contrast to the naked eye. One of the major problems here is that there is a minimal and no statistical difference in the community demographics of the two communities.

4. Methods: It is not clear from the methods what immature stages were being counted. Was it only 4th instar larvae and pupae? Also you mention Ae. albopictus in the methods but no mention after that. If no Ae. albopictus were found state it there.

5. In various places, proportions/percentages are given in ranges but it is not clear if this is between years or another spatial unit. There is mention of 21 neighborhoods, but I don’t think it applies here.

6. Note for the Container productivity, also refer to the table and simply describe broader trends. Something like: “Overall we collected 35% more containers and more than double pupae positive containers in rural sites compared to urban sites. Add a column for the per house estimates of the number of containers.

7. The Stegomyia indices reported in Figure 3 are quite high overall, even for the urban communities. Compare this to Line 254-257, when 18-46% of the houses had at least one container with larvae. Looking at the distribution might better capture this. Overall, I would spend more text describing Figure 3 than you do on container productivity.

8. Presentation of model results. Could large trends be presented – that is overall, rainy season had higher indices that during the dry season; rural higher than urban; then you can point out differences between years, but we are most interested in main effects.

9. Although I agree that the literature does emphasize urban transmission of dengue by its vector Ae. aegypti, the presence of this vector in rural locations is not new, especially in Asia. The western hemisphere is a bit different because the species was eradicated in the 1960-70s and probably became reestablished in urban areas first and now expanding to other areas.

Specific Comments

Line 162. Change to using Prokopack aspirators (these are not backpack aspirators).

Line 182: Suggest you make a table of the indice definitions.

Line 223-236. This would be better presented in a table (it is in supplementary material, but I don’t seem to have access), but again let the reader see the table and highlight the differences which there does not appear to be.

Line 254-261. Not clear what the take hope is, this observation is best show with a House Indext or Breteau index. We know that Ae. aegypti distributions are very clustered. How do your statements have implications for vector surveillance.

Line 268-270. This information should go into a table that contrasts urban versus rural communities, although it seems like the excess production in rural communities is mostly rain-filled containers, possibly more useless containers than in urban settings?

Line 276: Again clarify are we talking about L4s only.

Line 346. Don’t understand the title here nor the discussion.

Reviewer #2: (No Response)

PLOS authors have the option to publish the peer review history of their article (what does this mean?). If published, this will include your full peer review and any attached files.). If published, this will include your full peer review and any attached files.

.

Reviewer #1: No

Reviewer #2: No

**Figure resubmission:**
---

## [Decision Letter · Decision Letter 1]

23 Dec 2025

PNTD-D-25-00902R1

Abundance of the vector Aedes aegypti in urban and rural areas in Managua, Nicaragua

Dear Dr. Juarez,

Thank you for submitting your manuscript to PLOS Neglected Tropical Diseases. After careful consideration, we feel that it has merit but does not fully meet PLOS Neglected Tropical Diseases's publication criteria as it currently stands. Therefore, we invite you to submit a revised version of the manuscript that addresses the points raised during the review process.

Please submit your revised manuscript within by Jan 22 2026 11:59PM. If you will need more time than this to complete your revisions, please reply to this message or contact the journal office at plosntds@plos.org. Please include the following items when submitting your revised manuscript:

We look forward to receiving your revised manuscript.

Kind regards,

Geoffrey M. Attardo

Academic Editor

Audrey Lenhart

Section Editor

Shaden Kamhawi

co-Editor-in-Chief

Paul Brindley

co-Editor-in-Chief

**Reviewers' Comments:**

Reviewer's Responses to Questions

**Key Review Criteria Required for Acceptance?**

**Methods**

-Are the objectives of the study clearly articulated with a clear testable hypothesis stated?

-Is the study design appropriate to address the stated objectives?

-Is the population clearly described and appropriate for the hypothesis being tested?

-Is the sample size sufficient to ensure adequate power to address the hypothesis being tested?

-Were correct statistical analysis used to support conclusions?

-Are there concerns about ethical or regulatory requirements being met?

Reviewer #1: Overall, previous feedback has been considered and the missing study descriptions provided, although I do not see the supplemental information directly. Based on reviewer responses I'm satisfied.

Only minor point here is the larval counting. You have clarified in the reviewer responses that you counted all stages (not usually done for productivity) but I would explicitly state that since it is unusual. I would also like some feedback about what the distribution of 1st, 2nd, 3rd, and 4th instars were. Not the focus of the manuscript but when you say you counted all instars, one wants to know what the distribution is.

Reviewer #3: (No Response)

Reviewer #4: The field methods are robust. I have concerns with the statistical analyses.

Much of the manuscript relies on what I would consider global comparisons - chi-square tests of raw values/counts across a few high-level categories. The authors admit there is "significant" (significance is undefined) heterogeneity in larval/pupal collections among houses in both areas and in both seasons; they yet then implement statistical tests that assume independence and homogeneity of variance. For instance, the authors admit that some containers were sampled multiple times (repeated measures violate independence). I'm also not following how global indices (such as those listed in table 1 which list the total sample size as a denominator) can be compared statistically - are there perhaps subunits of collections within each district? If so, then perhaps statistical tests make sense - if not, then its simply a descriptive comparisons. So much per household information is washed out (for instance, lines 203 - 205 indicate zero values were removed), I really wonder that if such heterogeneity was actually analyzed, if the differences would stand between "rural" and "urban".

The linear vs. non-linear comparisons I think needs to be redone. The listed package gamm4 can compare linear and non-linear terms which I would recommend the authors consider. Then, AIC scores are more directly comparable and the notion of "thresholds" would be more statistically valid (pending non-linear models actually out perform linear models). I'd also consider BIC and RMSE type values be considered in model development/comparison. Also, some sort of assessment for zero-inflation is also likely needed given descriptions of collections provided by the authors.

Since the authors admit there is high heterogeneity of collections, raw GLMM/GAMM tables should be provided - I could not access the listed code/data availability (perhaps user error in broken links). Its also unclear if GLMMs represent any sort of model selection outcome (forward/backward/stepwise) or if they assess interactions or if they represent a full model of additive terms or individual unimodal glmm comparisons of each value. Finally, authors should state what the random terms were.

**Results**

-Does the analysis presented match the analysis plan?

-Are the results clearly and completely presented?

-Are the figures (Tables, Images) of sufficient quality for clarity?

Reviewer #1: Vast improvement, you might consider in Table 3 (adults also including adults/household - although you do describe this in the text).

Your efforts to streamline the manuscript have greatly paid off - much better manuscript.

Reviewer #3: (No Response)

Reviewer #4: The original reviewers identified a need to further identify how one region was urban and the other was rural. I would consider edits in response of this criticism insufficient. The authors mention they use categories defined by the MOH with rural sites being deficient in something - provided results clearly show the "rural" site as deficient but its never explicitely stated exactly how the rural region is distinguished from the urban. Seeing all of the data and amendments to the text, this feels more like a socio-economic gradient rather to a land cover gradient. Rural will be conflated with agrarian/farming and low density - all of the data presented suggest this is more of a comparison between urban/suburban type populations with some areas having infrastructure and other areas not having infrastructure. That would make more sense that aegypti is more prevalent in spaces that lack infrastructure. Perhaps that is less flashy of a result, but it would still be important to reinforce to a global audience - perhaps there is room for discussing the nuances of urban/rural classifications when it comes to a global vector?

**Conclusions**

-Are the conclusions supported by the data presented?

-Are the limitations of analysis clearly described?

-Do the authors discuss how these data can be helpful to advance our understanding of the topic under study?

-Is public health relevance addressed?

Reviewer #1: Again I am very satisfied with the authors efforts.

Reviewer #3: (No Response)

Reviewer #4: See methods/results comments

**Editorial and Data Presentation Modifications?**

Reviewer #1: Line 180. Prokopack aspirators instead of Prokopacks aspirators

Reviewer #3: (No Response)

Reviewer #4: Standard errors of the mean should be reported with averages, not standard deviations

Table 2. sometimes a period is used for large numbers rather than a comma. Plus, this again is also a global comparison/description. The authors mention high heterogeneity of containers at a per house level. something to consider adding to this table is number of houses inspected, number of houses with positive containers. etc. - things like that

Figure 3 - I'm not sure how confidence bars nor significance is determined considering these would be values calculated across the entirety of the data. I'd also consider re-orienting this figure temporally rather than by dry/wet.

Table 3: inside vs. outside? is this distinction made in any of the other analyses (for instance, is this considered in GLMMs/GAMMs). I have similar concerns for Figure 4 and 5.

Figure 6 is a plot of the smoothed function (not necessarily a prediction of the number of pupae compared to the number of adults). A prediction plot with observed data would be much more informative that this (which I would consider an internal diagnostic image of the smoothed function)

**Summary and General Comments**

Reviewer #1: I am very satisfied with the manuscript improvement and appreciate the authors efforts to address the reviewer concerns.

Minor issue which is completely optional. I'm opposed to the term "breeding sites" when describing larval habitats. Breeding is not happening in those sites, but larval development is. I recognize the term is common in the literature but consider making that change.

Reviewer #3: The authors have satisfactorily addressed the comments from the reviewers.

Reviewer #4: I think the original reviewers were correct to request more info on this being a rural vs. urban comparison. Global readers will think of this distinction as a function of economic drivers, when instead it appears more of a distinction of poverty. Much more context would be needed to explain this (pending the distinction even remaining if/when statistical modifications are made to further refine violations of independence at multiple levels within the data).

PLOS authors have the option to publish the peer review history of their article (what does this mean?). If published, this will include your full peer review and any attached files.). If published, this will include your full peer review and any attached files.

.

Reviewer #1: No

Reviewer #3: No

Reviewer #4: No

**Figure resubmission:**
---

## [Decision Letter · Decision Letter 2]

13 Apr 2026

Dear Dr. Juarez,

We are pleased to inform you that your manuscript 'Abundance of the vector Aedes aegypti in urban and rural areas in Managua, Nicaragua' has been provisionally accepted for publication in PLOS Neglected Tropical Diseases.

Best regards,

Nigel Beebe, PhD

Section Editor

Nigel Beebe

Section Editor

Shaden Kamhawi

co-Editor-in-Chief

Paul Brindley

co-Editor-in-Chief

Reviewer's Responses to Questions

**Key Review Criteria Required for Acceptance?**

**Methods**

-Are the objectives of the study clearly articulated with a clear testable hypothesis stated?

-Is the study design appropriate to address the stated objectives?

-Is the population clearly described and appropriate for the hypothesis being tested?

-Is the sample size sufficient to ensure adequate power to address the hypothesis being tested?

-Were correct statistical analysis used to support conclusions?

-Are there concerns about ethical or regulatory requirements being met?

Reviewer #4: (No Response)

**Results**

-Does the analysis presented match the analysis plan?

-Are the results clearly and completely presented?

-Are the figures (Tables, Images) of sufficient quality for clarity?

Reviewer #4: (No Response)

**Conclusions**

-Are the conclusions supported by the data presented?

-Are the limitations of analysis clearly described?

-Do the authors discuss how these data can be helpful to advance our understanding of the topic under study?

-Is public health relevance addressed?

Reviewer #4: (No Response)

**Editorial and Data Presentation Modifications?**

Reviewer #4: (No Response)

**Summary and General Comments**

Reviewer #4: (No Response)

PLOS authors have the option to publish the peer review history of their article (what does this mean?). If published, this will include your full peer review and any attached files.). If published, this will include your full peer review and any attached files.

.

Reviewer #4: No

---

## [Editor Report · Acceptance letter]

Dear Dr. Juarez,

We are delighted to inform you that your manuscript, "Abundance of the vector Aedes aegypti in urban and rural areas in Managua, Nicaragua," has been formally accepted for publication in PLOS Neglected Tropical Diseases.

Best regards,

Shaden Kamhawi

co-Editor-in-Chief

Paul Brindley

co-Editor-in-Chief
